# Rapid Weight Loss Coupled with Sport-Specific Training Impairs Heart Rate Recovery in Greco-Roman Wrestlers

Roberto Roklicer [1,*], Carlo Rossi [2], Antonino Bianco [2], Valdemar Štajer [1], Nemanja Maksimovic [1], Marko Manojlovic [1], Barbara Gilic [3,4], Tatjana Trivic [1] and Patrik Drid [1]

1    Faculty of Sport and Physical Education, University of Novi Sad, 21000 Novi Sad, Serbia; stajervaldemar@yahoo.com (V.Š.); nemanjamaksimovic1998@gmail.com (N.M.); markomanojlovic1995@gmail.com (M.M.); ttrivic@yahoo.com (T.T.); patrikdrid@gmail.com (P.D.)
2    Sport and Exercise Sciences Research Unit, University of Palermo, 90144 Palermo, Italy; rcarlo97@hotmail.com (C.R.); antonino.bianco@unipa.it (A.B.)
3    Faculty of Kinesiology, University of Split, 21000 Split, Croatia; barbaragilic@gmail.com
4    Faculty of Kinesiology, University of Zagreb, 10000 Zagreb, Croatia
*    Correspondence: roklicer.r@gmail.com

**Abstract:** Wrestling is a sport that can be classified with the use of alternating aerobic–anaerobic metabolism with moderate but high-impact energy expenditure. Heart rate recovery (HRR) is the difference between heart rate during exercise and a certain amount of time after the start of recovery. The goal of this study was to determine the difference in HRR between two phases: high-intensity sport-specific training (HISST) combined with rapid weight loss (RWL)—phase 1 (P1) and HISST only—phase 2 (P2). Ten national-level wrestlers were included in this study. All participants underwent HISST along with RWL procedures for P1. Seven days later, during P2, an identical training session was performed with no RWL included. We found a statistically significant difference in the values obtained after the first and second minutes of recovery in the second set for both cases ($p = 0.034$ and $p = 0.037$, respectively), with higher HR values recorded in P1. It can be concluded that there is undoubtedly a difference in HRR during training and RWL compared to HISST alone. Additionally, HISST along with RWL could compromise the aerobic component of recovery.

**Keywords:** aerobic recovery; wrestling; weight reduction; intermittent exercise

## 1. Introduction

Wrestling is widely recognized as one of the oldest sports in human history, and as such, it was performed within the original events in the ancient Olympic Games [1]. Currently, two styles of wrestling are included in the Olympics: Greco-Roman style, which strictly allows upper body techniques only, meaning that holds below the waist are forbidden, and is only practiced by men. The second is freestyle wrestling where athletes are allowed to use lower extremity techniques and trips and is practiced by both men and women across the world [2,3]. Both styles are characterized as vigorous and intermittent and belong to weight-categorized sports. Due to the nature of the fight, physical demands include explosive strength, aerobic endurance, and anaerobic capacity in terms of achieving competitive success [4].

Although combat sports (CS) are predominantly formed by intermittent short-term high-intensity actions, where anaerobic glycolysis is used as an energy resource as numerous efforts are employed, aerobic metabolism is also involved. Therefore, following a period of training sessions with intermittent efforts, there is an increase in the activity of the sympathetic nervous system observed after HRV analysis [5]. Accordingly, heart rate variability (HRV) is an important tool for monitoring the training process by controlling training load [5]. Some previous research suggests that aerobic capacity belongs to one of the most important physical factors for achieving good results in wrestling competitions [6].

The aerobic system in CS primarily contributes to the athlete's ability to sustain effort for the span of the combat and to recover during the brief periods of rest or decreased exertion [7,8]. Heart rate response during exercise and the associated physiological mechanisms involved have been extensively studied [9,10]. Certain studies imply that heart rate recovery (HRR) following exercise is a marker of training-induced alterations in autonomic control [11,12].

A decrease in heart rate (HR) values in the first minutes of recovery following physical activity has been proposed as an inexpensive, non-invasive, valid, and simple indicator of parasympathetic activity, which is nowadays broadly used in clinical practice [13–15]. However, it is also employed in the monitoring and prescription of athletic training [16], along with cardiovascular condition assessment [17]. In addition, it is widely believed that faster HRR is related to better athletic performance [16]. Rapid recovery of HR following moderate to heavy exercise may be an important mechanism for preventing excessive cardiac work. Athletes characterized by high aerobic capacity could be better adapted to maximal control of the heart rate reserve [18]. Thus, monitoring HR in specific wrestling training sessions can provide information on setting the appropriate intensities and be very useful for creating a training program.

Combat sports athletes commonly engage in rapid weight loss (RWL) procedures to compete in the upper spectrum of a lower weight category and likely gain a competitive edge over their rivals [19]. Typically, RWL is defined as a reduction of 5% of body weight less than one week before the competition [20,21]. Rapid weight loss methods are largely similar between combat sports and commonly include reduced fluid and food intake, heated training, rubber suits, sauna use, increased physical activity, etc. The interactive effect of food restriction and fluid deprivation might provoke adverse physiological effects on the body, leaving the athlete potentially unprepared for competition [22]. Additionally, there is evidence that RWL of 5% combined with strenuous exercise before competition causes a significant degree of muscle damage in CS athletes, thus making the competitive performance fairly questionable [23]. Since CS athletes usually have to achieve their target weight in a relatively short period, they usually have to go through strenuous training along with other RWL procedures. In this way, such training could stimulate increased sweating and thus facilitate weight loss. This crossover study deals with monitoring the HRR in a group of Greco-Roman wrestlers during high-intensity sport-specific training on two different occasions.

The objective of the study was to establish whether strenuous HISST along with RWL procedures affect the aerobic capacity in terms of HRR differently than HISST alone.

## 2. Materials and Methods

### 2.1. Participants

Ten male national-level Greco-Roman wrestlers participated in this study (n = 10; mean age: 22.44 ± 4.53 years; mean body weight: 73.36 ± 4.42 kg; mean body height: 174.43 ± 3.78 cm). At the time of the examination, all subjects were free of injury and voluntarily participated in the study by signing the informed consent. During the experiment, participants trained about 1–2 h daily following their regular training schedule. To be included in the study, each participant had to have a competitive experience of at least 5 years and have been performing rapid weight loss techniques over the last two years. Body composition parameters were measured using Omron weight scale BF511 (Omron, Japan). All athletes used increased exercise, skipping meals, and fluid restriction to reduce the desired 5% of body weight. The study was conducted in accordance with the Declaration of Helsinki.

### 2.2. Testing Protocol High-Intensity Sport-Specific Training (HISST)

This study included two phases. In phase one (P1), the group composed of combat sports athletes had to lose 5% of their body weight and carried out HISST while heart rate was monitored. In this phase, participants started losing weight rapidly two days before the day of the training. The day of training (testing) was the last day of their RWL period,

after which the measurements were performed. In the second phase (P2), seven days later, an identical high-intensity training protocol was carried out with no rapid weight loss procedure included (Figure 1). The total duration of the training was 90 min. All the participants were familiar with the testing procedure. The participants were instructed to perform the protocol with the same sparring partner on both occasions (P1 and P2). The high-intensity sport-specific training consisted of four sets of a maximal number of throws with work to rest ratio: 15:45 s, with each set lasting 10 min with a break of 3 min in between. Heart rate recovery was measured at three time points: the first, second, and third minute after each set of throws. During the test, participants performed shoulder throws. A heart rate monitor was attached to all participants (Polar-H10 heart rate sensor, Polar Pro chest strap) from the beginning to the end of the training session. The heart rate monitoring was performed using an iPad Pro 10, Polar Team application.

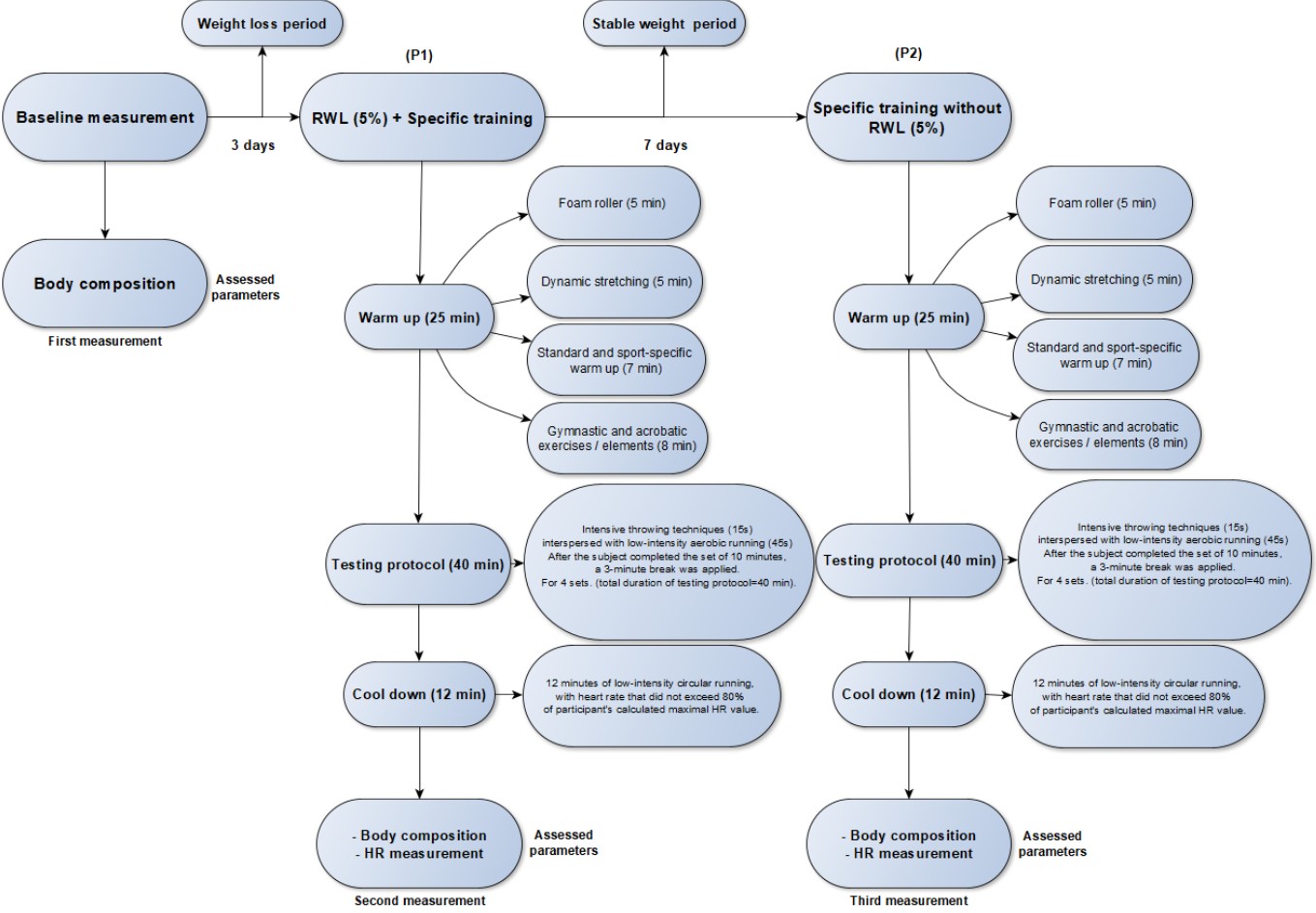

**Figure 1.** Experimental procedure design.

### 2.3. Warm-Up

Participants started with a warm-up that consisted of 5 min of foam rolling, followed by 5 min of dynamic stretching. Standard and sport-specific warm-ups were conducted for the next 7 min, after which 8 min of gymnastic and acrobatic elements were performed by wrestlers (total warm-up time = 25 min).

### 2.4. Testing Protocol

The main part of the training consisted of an intensive throwing technique (15 s) interspersed with low-intensity aerobic running (45 s). In fact, the participant initially began a low-intensity circular run (45 s). Five seconds before the throwing part of the test, the participant was instructed to place himself at a point 9 m away from the sparring

partner. At the sound signal, he had to run to his sparring partner as fast as possible and perform the throwing technique. Immediately after the throw, he had to return to the starting point of 9 m. Participants had to perform 4 shoulder throws together with sprints during this period of 15 s. When the throwing part was completed, participants started the next set of low-intensity running. After the subject completed the set of 10 min, a 3 min break was applied. Four sets of the explained protocol were completed (total duration of testing protocol = 40 min).

*2.5. Cooldown*

At the end of the main part of the training, a cooldown phase was employed. This phase consisted of 12 min of low-intensity circular running, with heart rate not exceeding 80% of participants' maximal calculated HR values.

*2.6. Statistical Analysis*

The results of the study are presented as mean values and standard deviations. For the normality of distribution, the Shapiro–Wilks test was performed. A paired sample *t*-test was used to compare the means between the two measurements using IBM SPSS Statistics for Windows, 20.0 (IBM Corp 20, Armonk, NY, USA). The statistical significance level was set at $p < 0.05$.

## 3. Results

The characteristics of combat sports athletes are presented in Table 1. The differences in body composition are visible across three time points: baseline values, P1, and P2 (Figure 2).

**Table 1.** Descriptive characteristics of wrestlers' body composition (n = 10).

| | Baseline Values | RWL and Training Phase (P1) | Training-Only Phase (P2) |
|---|---|---|---|
| Bodyweight (kg) | 73.36 ± 4.42 | 69.27 ± 4.12 * | 72.38 ± 4.17 # |
| BMI (kg/m$^2$) | 24.11 ± 0.96 | 22.62 ± 0.98 * | 23.64 ± 1.07 # |
| Fat mass percentage (%) | 16.37 ± 2.22 | 12.74 ± 3.15 * | 14.98 ± 2.47 # |
| Muscle mass percentage (%) | 42.51 ± 1.41 | 44.64 ± 2.14 * | 43.36 ± 1.71 # |
| Visceral body fat (%) | 6.11 ± 1.05 | 4.88 ± 1.16 * | 5.66 ± 1.32 # |
| Basal metabolic rate (kcal) | 1717.33 ± 64.98 | 1677.11 ± 61.49 | 1704.11 ± 56.43 |

Note: The values are presented as mean values ± standard deviation; * statistically significant difference in relation to the baseline values, $p \leq 0.001$; # statistically significant difference in relation to phase P1, $p < 0.01$.

Statistically significant differences were evident for all variables, except for the basal metabolic rate when baseline values were compared to P1 ($p \leq 0.001$). Additionally, significant differences were visible when P1 was compared to P2 ($p < 0.01$) for all examined variables except basal metabolic rate.

Heart rate recovery values for the two phases (P1 and P2) are presented in Figure 3. Greater values are observed in P1 for all performed sets except the second minute of recovery in the third set, which was slightly but not significantly lower ($p < 0.66$).

Statistically significant differences were observed in two recovery periods. These were evident in the first and second minute of recovery after the second set of the testing protocol. Higher values after the first minute of recovery in the second set during P1 were achieved in comparison with P2, 169.77 ± 8.94 vs. 158.22 ± 12.07 bpm, respectively ($p = 0.034$). Likewise, the second minute of recovery for the same set was also significantly higher in P1 compared to P2, 151.44 ± 11.92 vs. 143.88 ± 11.78 bpm, respectively ($p = 0.037$). A large difference was visible after the third minute of recovery in the third set, as higher values were achieved in P1 compared to P2; Nevertheless, it was not statistically significant ($p = 0.083$).

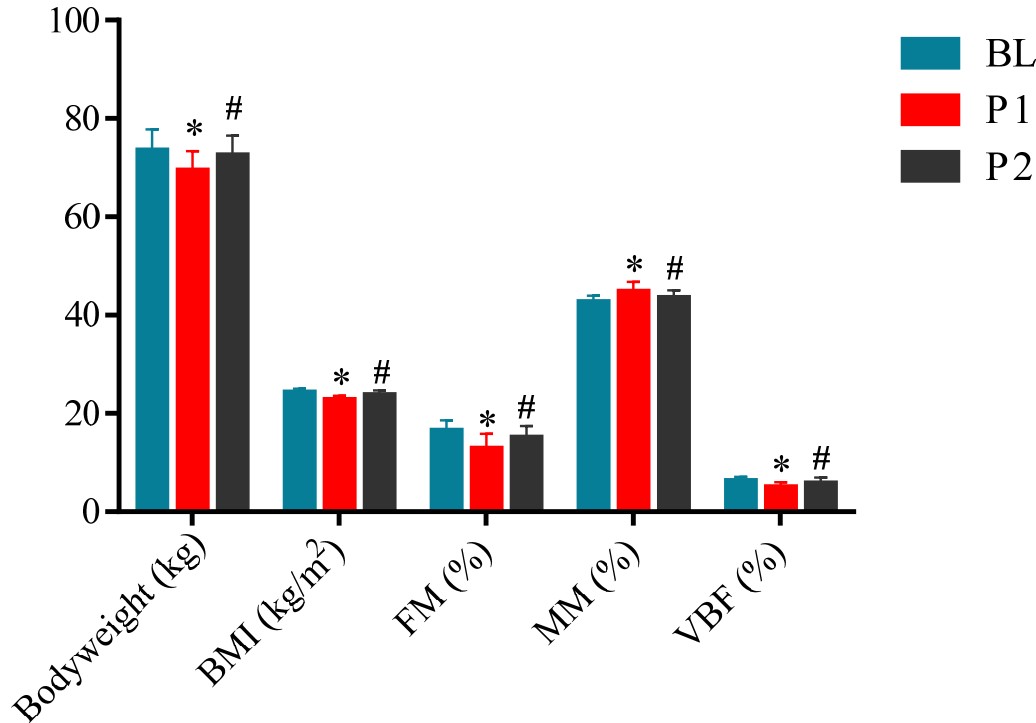

**Figure 2.** Descriptive characteristics of wrestlers' body composition. Legend: BL—baseline values; P1—phase one: HISST combined with RWL; P2—phase two: HISST with no RWL procedures included; BMI—body mass index (kg/m$^2$); FM—fat mass (%); MM—muscle mass (%); VBF—visceral body fat (%); * statistically significant difference in relation to the baseline values, $p \leq 0.001$; # statistically significant difference in relation to phase P1, $p < 0.01$.

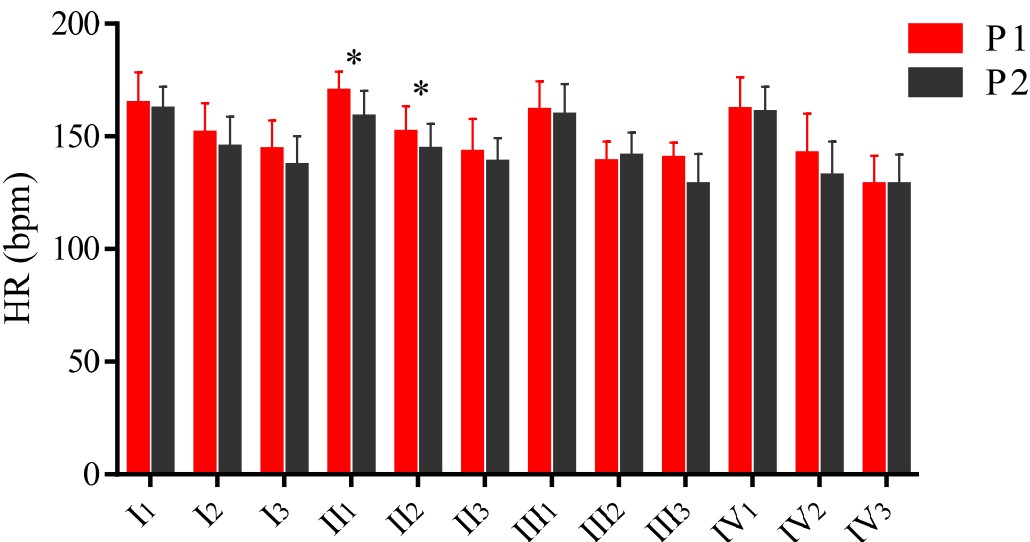

**Figure 3.** Heart rate recovery during 2 phases. Legend: I1—first minute of recovery in first round, I2—second minute of recovery in first round, etc.; P1—phase one: HISST combined with RWL; P2—phase two: HISST with no RWL procedures included; HR bpm—heart rate values; * statistically significant difference, $p < 0.05$.

## 4. Discussion

The objective of this study was to compare and evaluate the differences in HRR between HISST combined with RWL and HISST only, as participants went through identical training procedures during both phases of the experiment. The results of our study indicate a trend of higher values in P1 compared to P2. Such results demonstrate the lower ability of HRR when RWL procedures are included along with sport-specific training. The type of training and performance are additional components that can affect the HR response to recovery, as physiological adaptations of highly trained athletes can modify many aspects of exercise metabolism [24].

Heart rate variability is considered to be a helpful tool to monitor athletes' responses to training and recovery during competition, also including repeated anaerobic activities within and between matches [25]. Thus, it can be used for fitness evaluation of athletes and further prescription of the appropriate training program. Aerobic metabolism might be quite important in CS athletes to maintain the high intensity for the activities and assist with fast recovery of energy metabolism via resynthesis of creatine phosphate during the breaks and low-intensity activities [26,27]. Apparently, it seems that the ability to recover between aerobic efforts could determine the performance among CS athletes who are regularly engaged in high-intensity intermittent activity [25,28,29].

A study carried out by Ostojic et al. [24] examined HR recovery in ultra-short periods and found that athletes who engaged in intermittent endurance sports demonstrated faster recovery at 10 and 20 s following maximal effort compared to athletes competing in continuous sports. Specifically, the authors discovered that basketball, soccer, and handball players can recover faster compared to distance cyclists and runners. These results suggest that ultra-short heart rate recovery may be affected by the type of sports activity in well-trained athletes. Another study conducted by Ostojic et al. [18] on 32 healthy young soccer players that consisted of elite and sub-elite athletes assessed the HRR in ultra-short periods following an individualized ramp protocol to symptom-tolerated maximum using a treadmill system. The study revealed significantly lower HR values during the recovery period in elite compared to sub-elite players ($p < 0.05$) within the first 10 and 20 s. However, the first minute of HRR in elite soccer players can be compared with the values of wrestlers from our study during phase 1 (164 ± 5 and 164.2 ± 14.1 bpm, respectively). Another study conducted by Watson et al. [30] examined the short-term HRR in 84 collegiate male and female hockey and soccer players. These authors have found that HRR was faster at 10 and 30 s among intermittent sport athletes with a higher level of aerobic fitness. In this study, it was also stated that $VO_2max$ was an independent predictor of more rapid HRR at 10 s, but not anytime thereafter. They found no significant independent relationship between HRR and sex, body fat percentage, or lean body mass at any of the time points measured. A study conducted by Suzic Lazic et al. [16] measured the HRR among 137 elite athletes competing in soccer, basketball, and water polo. Heart rate recovery was examined during the first, second and third minute following a progressive maximal ramp protocol on a treadmill. When compared to the results of our study, the first minute of HRR was comparable only with the P2 values of our study for all four sets, and the third and fourth set of P1. Heart rate recovery values after the second minute of recovery can be compared only to the last set of P2 in wrestlers. Nevertheless, the third-minute HRR values were much lower in this group of team sport athletes in comparison with participants from our study.

Although the two test procedures might not be quite comparable, according to the classificatory table for the Special Judo Fitness Test (SJFT) created by Franchini et al. [31], participants from our study demonstrated poor HRR after the first minute (range 162–165 bpm). Since the recovery could be hindered by merging the HRR with HISST, RWL procedures should most likely be altered by CS athletes ahead of the official weigh-in. In practice, this means that wrestlers should consider using either typical RWL methods or high-intensity training to achieve the targeted weight category, otherwise, they might attend the competition insufficiently recovered from weight-loss procedures by combining the two. However,

athletes should employ less strict methods of weight loss before weigh-in. These results also suggest that HISST should be avoided before the competition.

This study has two limitations. The test procedure used in our study is not a standardized test protocol. However, it is created to simulate intensity during the in-season competitive period, including vigorous throws and sprints interspersed with low-intensity aerobic running. Additionally, the study was carried out on a relatively small number of participants. Further investigations should be conducted on a larger sample of athletes. To our knowledge, this extensive testing procedure and monitoring heart rate recovery for 40 min of the main training period (both in P1 and P2) was applied for the first time, which outlines the strength of the study.

## 5. Conclusions

Based on the results obtained, a trend of higher HR values during P1 was discovered, meaning remarkably higher HR values were present when RWL procedures were applied together with HISST. Current results might indicate a decrease in aerobic abilities in terms of HRR when RWL procedures are included along with sport-specific training. Wrestlers who choose to lose weight in this manner might not be adequately prepared for competition. Coaches and athletes around the world should be encouraged to avoid the combination of traditionally employed RWL techniques along with intensive training days prior to competition as it could impair athletes' recovery during and between matches.

**Author Contributions:** Conceptualization, R.R., T.T. and P.D.; Data curation, C.R., V.Š., N.M. and B.G.; Formal analysis, A.B. and T.T.; Funding acquisition, P.D.; Investigation, R.R. and A.B.; Methodology, A.B., B.G. and P.D.; Project administration, C.R. and M.M.; Resources, M.M. and P.D.; Software N.M.; Supervision, C.R. and V.Š.; Validation, V.Š. and M.M.; Visualization, N.M.; Writing—original draft, R.R. and B.G.; Writing—review and editing, T.T. All authors have read and agreed to the published version of the manuscript.

**Funding:** This research received no external funding.

**Institutional Review Board Statement:** The study was conducted in accordance with the Declaration of Helsinki and approved by the Ethics Committee of the University of Novi Sad, Faculty of Sports and Physical Education (protocol code. No. 46-06-02/2020-1).

**Informed Consent Statement:** Informed consent was obtained from all subjects involved in the study.

**Acknowledgments:** The authors are grateful to the athletes and coaches who participated in this research.

**Conflicts of Interest:** The authors declare no conflict of interest.

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
