# Peer review of "Rapid Weight Loss Coupled with Sport-Specific Training Impairs Heart Rate Recovery in Greco-Roman Wrestlers"

_applsci, doi:10.3390/app12073286_

Round 1
Reviewer 1 Report
Hello.
The article falls into both the field and addresses an interesting topic. The research presentation is well done, orderly and explicit. The argument and the introduction are solid and support the idea of research. The methods are clearly presented, as well as the content of the research and bibliographic sources that demonstrate solid documentation.
Congratulations. All the best!.
Author Response
Dear reviewer, thank you for your comments.

Reviewer 2 Report
- Please provide figure 1 in better quality
- Please provide a bar graph of data present in table 1, only the statistically significant.
- Please describe why the II1 is different and not the II3 and IV3. They have almost the same difference between the two groups (Figure 2).
- Please insert the standard error in the graph.
Author Response
- Thank you for your comment. The quality of the figure 1 is now improved.
- Thank you for your comment. Graph for this table is now provided with the significant differences presented.
- Thank you for your comment. However, after the first minute of recovery in the 2nd round the statistically significant difference was observed, while after the third minute for the same round the difference was still visible without statistical significance. Additionally, after the third minute of the 4th round the mean values for HRR were the same.
- Thank you for your comment. The standard error is now added and is visible in the figure 3.

Round 2
Reviewer 2 Report
No comments needed